# The Burnout of Nurses in Intensive Care Units and the Impact of the SARS-CoV-2 Pandemic: A Scoping Review

Andreia Lima [1,*], Maria Teresa Moreira [2], Carla Fernandes [3], Maria Salomé Ferreira [4], Margarida Ferreira [5], Joana Teixeira [6], Mafalda Silva [5], Vítor Parola [7] and Adriana Coelho [7]

1 Research Center for Health Technologies and Services (CINTESIS@RISE), Institute of Research, Innovation and Development Fernando Pessoa Foundation, Faculty of Medicine, University of Porto (FMUP), 4200-319 Porto, Portugal
2 Research Center for Health Technologies and Services (CINTESIS@RISE), Institute of Research, Innovation and Development Fernando Pessoa Foundation, 4200-253 Porto, Portugal
3 Research Center for Health Technologies and Services (CINTESIS@RISE), Porto Higher School of Nursing, 4200-072 Porto, Portugal
4 The Health Sciences Research Unit: Nursing (UICISA: E), Polytechnic Institute of Viana do Castelo (IPVC), 4900-347 Viana do Castelo, Portugal
5 Research Center for Health Technologies and Services (CINTESIS@RISE), Health Sciences School Jean Piaget Vila Nova de Gaia, 4405-678 Vila Nova de Gaia, Portugal
6 Health Sciences School, University Fernando Pessoa, 4200-253 Porto, Portugal
7 The Health Sciences Research Unit: Nursing (UICISA: E), Nursing School of Coimbra (ESEnfC), Portugal Centre for Evidence-Based Practice: A Joanna Briggs Institute Centre of Excellence, 3000-232 Coimbra, Portugal
* Correspondence: amlima@ufp.edu.pt

**Abstract:** Background: The world's population changed with the emergence of the SARS-CoV-2 pandemic. Burnout arises due to overwork, prolonged work periods, a lack of human and material resources, etc. Several studies have reported the incidence of burnout syndrome in nurses that work in intensive care units (ICUs). The aim was to map the scientific evidence related to nurses' burnout in the ICU, namely the repercussions of SARS-CoV-2 in terms of burnout among nurses. Methods: A scoping review followed the Joanna Briggs Institute methodology guidelines to search for and synthesise studies published between 2019 and 2022. The databases searched were MEDLINE, CINAHL, LILACS, SCOPUS, PsycINFO and OPEN GREY. A total of fourteen articles were eligible to be included. Results: A content analysis of the selected articles was carried out, and three categories emerged that corresponded to the dimensions of burnout according to Maslach and Leiter: emotional exhaustion, depersonalisation dimension and a lack of personal accomplishment. It was evident that nurses who worked in the ICU during the pandemic showed high levels of burnout. Conclusions: It is recommended that hospital administrations hire health professionals, namely nurses, as a strategic and operational management strategy to reduce the risk of increased burnout during pandemic outbreaks.

**Keywords:** burnout; professional; nurses; intensive care units; SARS-CoV-2; review

## 1. Introduction

Intensive care medicine is a multidisciplinary and differentiated area that encompasses prevention, diagnosis and treatment in situations of severe acute illnesses with the potential to be reversed and which compromise one or more vital functions [1]. Thus, the intensive care units (ICUs) have as their primary objective the provision of life-sustaining care for people in critical situations [2,3]. This essential condition, combined with clinical instability, the need for monitoring, ethical dilemmas related to patient care, the atmosphere of tension, the high number of interventions, and the risk of error, mortality and morbidity, places everyone at risk of emerging multidisciplinary team challenges [4,5]. As such, all these

factors induce high levels of stress and physical and psychological fatigue and may result in exhaustion [6].

To achieve the nursing interventions necessary to assist people in critical situations, ICU nurses provide highly specialised care using state-of-the-art technology. In this context, the number of needed and desirable nurses to correspond directly to the requirements of the people cared for is challenging to delineate. Thus, given all these demands, professionals must constantly pursue adaptive and coping measures to maintain their health in all aspects, namely their mental health through emotional management [5].

According to Maslach and Leiter, burnout results from overwork, prolonged work periods, low wages, interprofessional conflicts, work overload, a lack of human and material resources and even professional disappointment [7,8]. Several studies report the incidence of burnout syndrome in nurses who develop their practices in the ICU [9,10], even corroborating that this is more frequent in nurses than in other multidisciplinary team professionals [11].

Burnout can develop from exposure to stressors and is characterised by emotional exhaustion, a decreased sense of personal achievement and depersonalisation. The American psychologist Herbert Freudenberger first used this concept to describe the results of prolonged exposure to stress and anxiety [12]. Nurses in the context of the ICU are subject to physical and psychological overload, resulting in insomnia, physical fatigue, irritability and anxiety, and may culminate in pathologies of a depressive nature [6,13], bordering on burnout.

In 2020, the world's population changed with the emergence of the SARS-CoV-2 pandemic, which caused profound changes of political origin in health organisations that influenced humanised care, with particular emphasis on the ICU, since it was in this context that most severe cases occurred and remained [14].

Given the above and since ICU nurses commonly face various challenges in their daily lives, the SARS-CoV-2 pandemic brought more significant concerns for these professionals. This global event placed them in situations of insecurity, participating in discussions about the end of life or prolonging it through artificial assistance, and at risk of providing inadequate care resulting from poor working conditions, consequently promoting changes in their physical and mental health. [15,16]. Thus, the design of strategies aimed at nurses' health, focused on burnout during the SARS-CoV-2 pandemic, is of concern.

A preliminary literature review was performed on PROSPERO, MEDLINE, Open Science Framework (OSF), Cochrane Database of Systematic Reviews and Joanna Briggs Institute (JBI) Evidence Synthesis, which revealed the need for a scoping review of the subject under study.

This scoping review aims to map the scientific evidence related to nurses' burnout in the ICU during the pandemic, namely the repercussions of SARS-CoV-2 in terms of burnout among nurses. The specific question of this research is: What is the impact of the SARS-CoV-2 pandemic on the burnout of nurses working in the ICU?

## 2. Materials and Methods

This scoping review followed the guidelines of the Joanna Briggs Institute (JBI) methodology [1,2]. The items identified in the reports prepared for the guidance of systematic reviews and extension of meta-analyses (PRISMA-ScR) were used [3]. The protocol was registered in the OSF (https://osf.io/8s7a6/ (accessed on 22 April 2022)): DOI 10.17605/OSF.IO/8S7A6. A previous protocol for this scoping review was previously published [14].

*2.1. Inclusion and Exclusion Criteria*

The following inclusion criteria, based on JBI recommendations that use the mnemonic PCC for scoping reviews, were: participants—this review considered studies that included nurses; regarding the concept—this review had studies addressing the burnout of nurses; concerning the context—this review contained articles in which the study period included articles that occurred during the SARS-CoV-2 pandemic and in ICUs, regardless of the country of the study; and types of sources—this scoping review considered all study typologies, i.e., quantitative, qualitative and mixed methods, and all kinds of literature reviews for analyses. All articles that did not respond to the aim of the study were excluded, specifically articles that did not address burnout in nurses in the ICU in the context of the COVID-19 pandemic.

*2.2. Search Strategy*

Considering the Peer Review of the Electronic Search Strategies (PRESS) checklist, two reviewers developed the search strategy, which was peer-reviewed by the third expert. The following databases were used to research primary and secondary, published and not published studies included in the review: Medical Literature Analysis and Retrieval System Online (MEDLINE®) via PubMed, and Cumulative Index to Nursing and Allied Health Literature (CINAHL®), LILACS, SCOPUS, PsycINFO and OPEN GRAY. We limited the studies' languages to the ones understood by the authors: English, Portuguese and Spanish.

The strategy research recommended by JBI was executed [17,18]. A preliminary search was carried out in the MEDLINE (via PubMed) and CINAHL Complete (EBSCOhost) databases to identify the keywords and index terms used in the articles. The search strategy was created for each database based on the findings. This search was conducted on 14 April 2022. The reference lists of all included articles were reviewed for the possibility of the inclusion of additional articles. The identified themes were added to the ENDNOTE program X9.3.1 (Clarivate, Chandler, AZ, USA) and RAYYAN and duplicates were removed. The search strategy used in this scoping review can be seen in Table 1.

All articles were retrieved for relevance according to the title and abstract. The full text of the selected articles was evaluated in detail alongside the inclusion criteria by the two leading independent reviewers. The data were obtained from the reports and included in the review individually by two reviewers using the data extraction form. The data extracted had: the author, year and country of the study, aims, population and sample size, the impact of burnout, and factors that contribute to the development of burnout in ICUs during the COVID-19 pandemic in the results; and also extracted the level of burnout of the nurses. Any disagreements regarding the inclusion of an article or data relevant for extraction were solved through discussion or with a third reviewer.

**Table 1.** Database search strategy and results.

Database: CINAHL Complete (via EBSCO)
Filters: last 4 years, English, Portuguese, Spanish, excluding MEDLINE
Results: 22
Search strategy (14 April 2022)
(TI SARS-CoV-2 OR AB SARS-CoV-2 OR MH SARS-CoV-2 OR TI COVID-19 OR AB COVID-19 OR MH COVID-19) AND (TI burnout OR AB burnout OR MH burnout, professional OR TI exhaustion OR AB exhaustion) AND (TI nurse OR AB nurse OR MH nurses) AND (TI intensive care units OR AB intensive care units OR MH intensive care units OR TI intensive care unit OR AB intensive care unit OR TI ICU OR AB ICU)

**Table 1.** *Cont.*

| |
|---|
| Database: Psychology and Behavioral Sciences Collection<br>Filters: last 4 years, English, Portuguese, Spanish<br>Results: 2<br>Search strategy (14 April 2022)<br>(((DE "PSYCHOLOGICAL burnout") OR (burnout) OR (exhaustion)) AND ((DE "NURSES") OR (nurse) OR (nurses) OR (DE "NURSE practitioners")) AND ((DE "INTENSIVE care units") OR (intensive care units) OR (intensive care unit) OR (ICU)) AND ((DE "SARS-CoV-2") OR (SARS-CoV-2) OR (DE "COVID-19") OR (COVID-19))) |
| Database: LILACS<br>Filters: last 4 years, English, Portuguese, Spanish<br>Results: 10<br>Research Strategy (14 April 2022)<br>(COVID AND burnout AND nurse) |
| Database: SCOPUS<br>Filters: last 4 years, English, Portuguese, Spanish<br>Results: 208<br>Search strategy (14 April 2022)<br>((TITLE-ABS-KEY ((burnout, professional) OR (burnout) OR (exhaustion)) AND TITLE-ABS-KEY ((nurses) OR (nurse) OR (nurses) OR (nurse practitioners)) AND TITLE-ABS -KEY ((intensive care units) OR (intensive care units)) OR (intensive care unit)) OR (ICU)) AND TITLE-ABS-KEY ((SARS-CoV-2) OR (COVID-19))) |
| Database: MEDLINE (via PubMed)<br>Filters: last 4 years, English, Portuguese, Spanish<br>Results: 50<br>Search strategy (14 April 2022)<br>(((((burnout, professional[MeSH Terms]) OR (burnout[Title/Abstract])) OR (exhaustion[Title/Abstract])) AND (((((nurses[MeSH Terms]) OR (nurse[Title/Abstract])) OR (nurses[Title/Abstract])) OR (nurse practitioners[MeSH Terms]))) AND ((((intensive care units[MeSH Terms]) OR (intensive care units[Title/Abstract])) OR (intensive care unit[Title/Abstract])) OR (ICU[Title/Abstract]))) AND ((((SARS-CoV-2[MeSH Terms]) OR (SARS-CoV-2[Title/Abstract])) OR (COVID-19[MeSH Terms])) OR (COVID-19[Title/Abstract])) |
| Database: OPEN GRAY<br>Results: 5<br>Search strategy (14 April 2022)<br>(burnout AND COVID-19) |

## 3. Results

### 3.1. Study Characteristics, Settings, and Sample

A total of 229 articles relevant to the present study were identified (after removing duplicates). After reading the title and abstract, a total of 189 articles were excluded by the assessment of two independent reviewers since they did not specifically address the ICU wards but rather the emergency departments; these studies also did not specify the sample size (n) of nurses but reported results of all health professionals. The full versions of the remaining 40 articles were read (1 could not be retrieved), and 14 were selected according to the inclusion criteria, as shown in Figure 1. Divergences appeared between the two independent reviewers in five articles, so the third reviewer analysed them. After this phase, the results from the eligibility criteria showed that 14 articles were published between 2020 and 2022.

The studies took place in 11 different countries, namely: the USA (n = 2), Belgium (n = 2), Brazil (n = 2), the UK (n = 1), Japan (n = 1), Russia (n = 1), Italy (n = 1), Taiwan (n = 1), South Korea (n = 1), Spain (n = 1) and Turkey (n = 1) in intensive care units (Table 2).

**Table 2.** Summary of articles' findings.

| Author, Year and Country | Aims | Population and Sample Size | Results (Level of Burnout) |
|---|---|---|---|
| Guttormson, et al., 2022, USA [19] | To describe the experiences of ICU nurses during the COVID-19 pandemic in the United States. | Nurses sample size (*n* = 285) | Nurses reported stress related to a lack of evidence-based treatment, poor patient prognosis, and lack of family presence in the ICU. Nurses perceived inadequate leadership support and inequity within the healthcare team. Lack of consistent community support to slow the spread of COVID-19 or recognition that COVID-19 was real increased nurses' feelings of isolation. Nurses reported physical and emotional symptoms, including exhaustion, anxiety, sleeplessness, and moral distress. Fear of contracting COVID-19 or of infecting family and friends was also prevalent. |
| Haruna et al., 2022, Japan [20] | To verify if lower mutual support among ICU HCP professionals is associated with increased probability of burnout. | HCW (*n* = 335) Subsample Nurses (*n* = 196) | The majority of respondents were nurses (58.5%), followed by physicians (18.5%). Nearly half of the respondents (53.7%) were men, with more than 10 years of ICU experience (80.9%). Approximately 85% of respondents were involved in COVID-19 patient management (85.4%). Men had a significantly lower probability of burnout than women ($p = 0.021$). Respondents with housemates had a significantly lower probability of burnout. ($p = 0.049$). Nurses had a significantly higher frequency of burnout than other healthcare professionals ($p = 0.044$). |
| Sevic et al., 2021, Turkey [21] | To measure the levels of anxiety and burnout among healthcare workers during the COVID-19 pandemic. | HCW (*n* = 104) Subsample nurses (*n* = 43) | The distributions of emotional exhaustion (EE), personal accomplishment (PA), and depersonalisation (DP) scores were significantly different among HCWs ($p = 0.023$, 0.000, and 0.000, respectively). Residents and nurses had almost the same EE scores ($p = 0.872$), both of which turned out to be higher than that for attending physicians ($p = 0.007$ and 0.003, respectively). The PA score for residents was lower than that for attending physicians ($p = 0.039$) and nurses ($p = 0.000$). |
| Seluch et al., 2021, Russia [22] | To establish the features of emotional burnout syndrome and its connection with typological characteristics of the personality in nurses working with COVID-19 patients. | *n* = 74 Outpatients = 30 ICU = 44 | Emotional burnout level: (24.57 ± 1.46) scores, the average value of DP level was (7.85 ± 0.8) points. Among nurses, 15 (34.1%) had a high level of EE, 23 (52.3%) had an average level, and only 6 (13.6%) had a low level. Regarding DP, among intensive care nurses, a high level of DP was seen in 5 (11.4%) people, medium in 30 (68.2%) people, and low in 9 people (20.4%). A high level of reduction in professional success was found in 3 (6.8%) intensive care nurses, an average level in 17 (38.6%), and a low level in 24 (54.6%). |

**Table 2.** *Cont.*

| Author, Year and Country | Aims | Population and Sample Size | Results (Level of Burnout) |
|---|---|---|---|
| Gordon et al., 2021, USA [23] | To examine ICU nurse's experiences caring for COVID-19 Patients. | *n* = 11 | ICU nurses for patients diagnosed with COVID-19 were categorised into five themes and subthemes. Emotions experienced were subcategorised into anxiety/stress, fear, helplessness, worry, and empathy. Physical symptoms were subcategorised into sleep disturbances, headaches, discomfort, exhaustion, and breathlessness. Care environment challenges were subcategorised into nurses as surrogates, inability to provide comforting human connection, dying patients, personal protective equipment (PPE), isolation, care delay, changing practice guidelines, and language barrier. Social effects were subcategorised into stigma, divergent healthcare hero perception, additional responsibilities, strained interactions with others, and isolation/loneliness. Short term coping strategies were subcategorised into co-worker support, family support, distractions, mind/body wellness, and spiritualty/faith. Results showed that ICU nurses experienced intense psychological and physical effects as a result of caring for patients diagnosed with COVID-19 in a challenging care environment. Outside of work, nurses faced pandemic-induced societal changes and divergent public perceptions of them. |
| Vitale el al., 2020, Italy [24] | To evaluate the burnout syndrome among nurses who are engaged in the care of patients with Coronavirus disease (COVID-19) | *n* = 291 | Regarding EE, female nurses were more exposed to the phenomenon than men ($p < 0.001$). However, for the other two dimensions, there were no statistically different differences between the two sexes (DP: $p = 0.809$; personal accomplishment: $p = 0.268$). It was seen that EE was significantly higher among female nurses with a range of years of work experience ranging from 0 to 10 ($p = 0.005$) compared to male nurses. Significant levels of EE were also highlighted considering female nurses, both those who were already assigned to an intensive care unit before the COVID-19 pandemic ($p = 0.003$) and those who were transferred during the health emergency ($p = 0.028$). |
| Srinivas et al., 2021, UK [25] | To identify the prevalence of burnout and the contributing factors amongst HCPs caring for COVID-19 patients admitted to ICU. | HCW (*n* = 153) Subsample nurses (*n* = 47) | The response rate was 79%. Nurses and other staff reassigned to work in the ICU had higher levels of burnout. Working in personal protective equipment was most distressing, followed by direct patient care. There were positive outcomes including learning opportunities, professional development and job satisfaction. The impact of the pandemic on staff burnout may have been mitigated by acknowledging the contribution of staff, improving communication and encouraging their to access support. |

**Table 2.** *Cont.*

| Author, Year and Country | Aims | Population and Sample Size | Results (Level of Burnout) |
|---|---|---|---|
| Molina-Mula, 2021, Spain [26] | To analyse the levels of anxiety, depression, PTS and burnout of nurses during the pandemic to identify possible sociodemographic and related occupational factors. | $n = 280$<br>ICU nurses ($n = 23$) | Nurses working in the COVID-19 ICU had a mean score of 31.67 ($p < 0.001$). The results showed statistically significant variations in emotional fatigue according to the unit, years of professional experience, health centre and experience in COVID-19 units ($p < 0.001$; F ANOVA, chi square). The findings indicated that after two months of working in units with COVID-19 patients, nurses began to experience emotional fatigue, which increased over the course of a month. The high fatigue values noted among nurses working in the adult ICU and in ICUs with COVID-19 patients are of concern; they exceeded the mean of 31 points out of 26, which is the detection cut off for fatigue. |
| Butera et al., 2021, Belgium [27] | Assess (1) the prevalence of burnout risk among nurses working in intensive care unitsand emergency department before and during the coronavirus disease 2019 pandemic and (2) the individual and work-related associated factors. | $n = 422$ (first-wave measurement)$n = 1616$ (second-wave measurement) | The overall prevalence of burnout risk increased significantly among intensive care unit nurses (from 51.2% to 66.7%, $x^2 = 23.64$, $p < 0.003$). During the pandemic, changes in workload and the lack of personal protective equipment were significantly associated with a higher likelihood of burnout risk, whereas social support from colleagues and from superiors and management was associated with a lower likelihood of burnout risk. Several determinants of burnout risk were different between intensive care unit and emergency nurses. |
| Ribeiro et al., 2021, Brazil [28] | To review the scientific literature on burnout syndrome among intensive care unit nurses during the COVID-19 pandemic | N/A | This narrative review considered publications on the current pandemic, as well as studies on worker health and burnout, focusing on intensive care unit nurses.The literature was organised into two thematic categories: (1) emotional distress in the daily work of intensive care unit nurses; (2) preventing burnout in these professionals during the COVID-19 pandemic. Although the literature on burnout is expressive, there is a need to transmit data produced during the pandemic and follow these professionals longitudinally, which could lead to the development of specific prevention and health promotion strategies. Changes in the emotional and working conditions of these professionals must become a permanent part of worker health research and practice, rather than a temporary measure during the pandemic. |

| Author, Year and Country | Aims | Population and Sample Size | Results (Level of Burnout) |
|---|---|---|---|
| Freitas et al., 2021, Brazil [29] | To evaluate the prevalence and existence of predictive factors for Burnout syndrome in nursing technicians who work in an intensive care unit during the COVID-19 pandemic. | *n* = 94 | The prevalence of the syndrome was observed in 25.5% of the analysed sample. The variables that, after multiple analyses, were shown to be predictors associated with a higher prevalence of Burnout syndrome were: age > 36 years, overtime considering the rigid workload and being elitist. |
| Bruyneel et al., 2021, Belgium [30] | To assess the prevalence of burnout risk and identify risk factors among ICU nurses during the COVID-19 pandemic. | *n* = 1135 | The overall prevalence of burnout risk was 68%. A total of 29% of ICU nurses were at risk of DP, 31% of reduced PA, and 38% of EE. A 1:3 nurse-to-patient ratio increased the risk of EE (OR = 1.77, 95% CI: 1.07–2.95) and SD (OR = 1.38, 95% CI: 1.09–2.40). Those who reported having a higher perceived workload during the COVID-19 pandemic were at higher risk for all dimensions of burnout. Shortage of personal protective equipment increased the risk of EE (OR = 1.78, 95% CI: 1.35– 3.34) and nurses who had reported symptoms of COVID-19 without being tested were at higher risk of EE (OR = 1.40, 95% CI: 1.68–1.87). |
| Chen et al., 2021, Taiwan [31] | Report the effects of COVID-19 on healthcare workers' trauma andburnout and to analyse the factors associated with mental health outcomes | *n* = 12,596 | The proportion of participants who experienced high levels of DP was significantly higher for men (Women: 17.9%; Men: 22.3%) and those who worked in critical care units (Yes: 21.1%; No: 16.9%) and departments related to COVID-19 (Yes: 22.7%; No: 17.1%). The average score in lack of PA score was 19.0 8.4, indicating that the participants experienced a lack of PA to a small degree. |
| Jang et al., 2021, Korea [32] | To evaluate the current status of emotional exhaustion and peritraumatic distress in healthcareworkers (HCWs) in the COVID-19 pandemic, and identify factors associated with their mental health status | *n* = 1112 *n* nurses = 655 | Although no significant difference in peritraumatic distress was observed among the surveyed HCWs, the workers' experiences of EE varied according to work characteristics. Responders who were female, older, living with a spouse, and/or full-time workers reported higher levels of EE. Public health officers and other medical personnel who did not have direct contact with confirmed patients and full-time workers had a higher level of peritraumatic distress. Forced involvement in work related to COVID-19, worry about stigma, worry about becoming infected, and perceived sufficiency of organisational support negatively predicted EE and peritraumatic distress. |

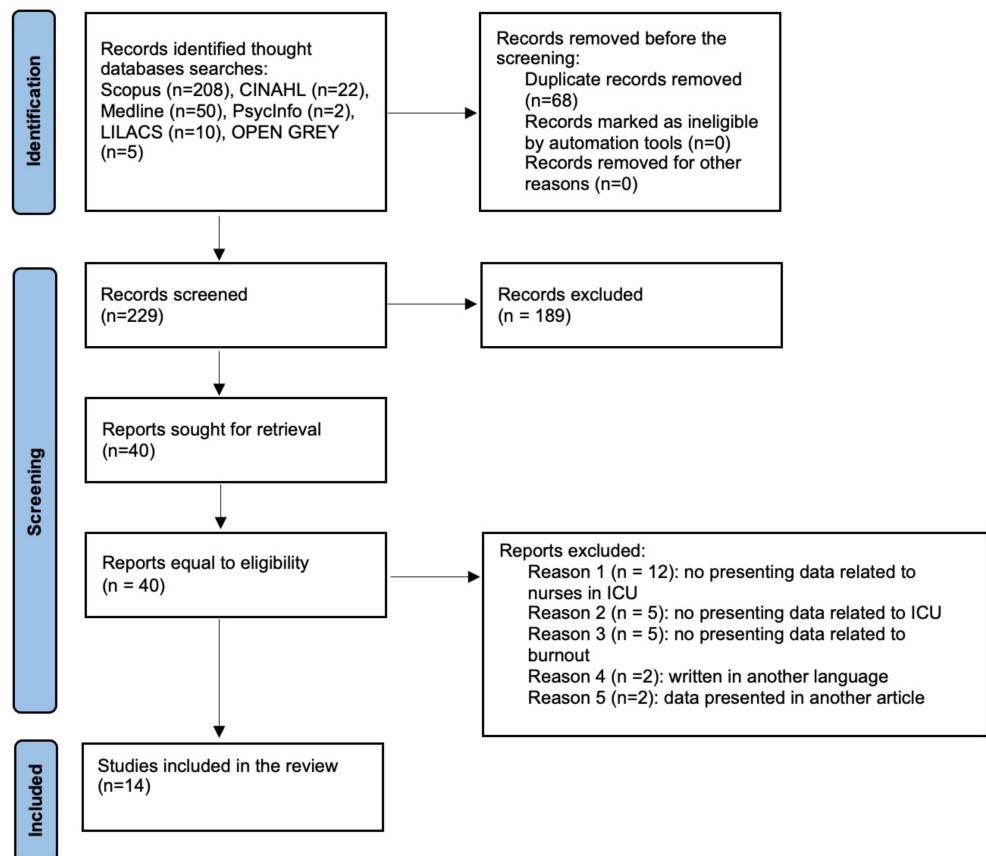

**Figure 1.** PRISMA flowchart of the study selection process.

*3.2. Burnout in Intensive Care Units*

According to some studies [19,23], nurses present a higher level of burnout among the health professionals who work in the ICU. Among these, the female gender is the one that shows the highest level, being also more prevalent for those who exercise a professional activity in the first ten years [23].

A content analysis of the selected articles was carried out, and three categories emerged that correspond to the dimensions of burnout according to Maslach and Leiter [7]: emotional exhaustion, depersonalisation dimension and a lack of personal accomplishment.

3.2.1. Emotional Exhaustion

The situation generated by the COVID-19 pandemic forced families to separate. This event was very distressing for nurses, as many left their homes and went to live in other places where they did not risk contamination [19,23].

On the other hand, in the work context, these professionals experienced distressing situations resulting from the consecutive experience with death and the loneliness in the dying process that the pandemic enforced on others [23]. During the pandemic, nurses played a fundamental role in monitoring the final phase of life since it was necessary to assume the role of monitoring death [23], sometimes limited by the lack of resources due to the increased needs to fulfil this role by others [31]. Study participants [23] highlighted in their responses that the most challenging thing was holding the hand of a dying person while their family was outside the institution crying, generating in them a feeling of impotence, despair and great dissatisfaction.

The authors [20] reported that nurses' emotional exhaustion level in the ICU during the COVID-19 pandemic was moderate, with 34.1% having a high level of emotional exhaustion, 2.3% a medium level and 13.5% a low level. Nurses who had worked in the ICU for a long time had higher dynamic exhaustion values than those allocated to the ICU during the COVID-19 pandemic [23].

The need to use personal protective equipment (PPE), combined with the physical overload of life-threatening care while providing care to patients who were victims of COVID-19, caused exhaustion, shortness of breath and discomfort in professionals, especially nurses [22,23,26,30].

### 3.2.2. Depersonalisation

The pandemic led to questioning one's choice of profession, leading some ICU nurses to ask whether they wanted to continue to practice nursing. This was a very challenging moment for nursing, allowing nurses to work on their resilience [23].

According to the authors [20], 11.4% of the nurses who worked during the COVID-19 pandemic in the ICU had a high level of depersonalisation, 68.2% a medium level and 20.4% a low level. In general, higher levels of depersonalisation were observed in men with professional experience between 11 and 20 years, compared to women with the same amount of professional experience [23].

Study participants [22] indicated that in providing care during the COVID-19 pandemic, feelings of inability to establish an empathic relationship, isolation, many death experiences, concerns about PPE, delay in providing care, changes in practical guidelines every minute and language barriers were factors that led to distancing themselves from others and work. These characteristics are present in the definition of depersonalisation in burnout.

### 3.2.3. Personal Achievement

Due to the increase in the mortality rate generated by the COVID-19 pandemic, nurses presented feelings of frustration in their responses since, despite all the efforts they instilled to save lives, they still witnessed daily the slow death of the people they cared for [22].

The authors [20] in their study showed that 6.8% of the nurses who performed functions during the COVID-19 pandemic in the ICU had a high level of low personal accomplishment, 38.6% had a medium level and 54.6% had a low level.

In the study [23], when presenting the low personal accomplishment subscale score for females and males, it was presented at a moderate level, with scores of 32, 8 and 31.7, respectively.

## 4. Discussion

Nurses working in the ICU have a high workload associated with cognitive demands due to the complexity of the technology used, which can cause mental and physical overload due to the critical situation of the patients and the imminent risk of death. In this context, nurses must provide more complex nursing care, leading to increased attention and immediate decision making. The literature also shows that nurses working in the ICU experience high stress levels, which manifest as psychological and physical effects [19,23]. The American Nurses Association (ANA, 2020) highlighted in its study, in which 10,000 nurses participated, that 50% of the sample reported feeling overwhelmed, 30% had symptoms of depression, and more than 70% had sleep disorders.

In general, it was found that nurses in the ICU reported symptoms of burnout during the COVID-19 pandemic, where high levels of emotional exhaustion and low professional efficacy were reported [19,20,22,24,27,29,31,32]. We can perceive that the percentage of ICU nurses with symptoms is considerably higher than those reported before the COVID-19 pandemic. It is, therefore, reasonable to assume that overwork, combined with the demands of specific work-related changes caused by the COVID-19 pandemic, has led to a significant increase in burnout symptoms [19–21,25,30–32]. This event may be associated not only with overwork due to the rise in the number of patients in critical condition but also with fear and apprehension regarding the novelty of the pathology and its repercussions.

It was evident that nurses who worked in the ICU during the pandemic displayed high emotional exhaustion, having reported feelings of reduced professional efficacy and depersonalisation concerning their professional work [19,21,22,24,27,30–32]. This is not

an unexpected corroboration since the literature demonstrates that nurses in the ICU are at greater risk due to their position within the work organisation and the tasks assigned to them. They are always on the front line in times of crisis. The increase in pressure combined with clinical demands during the COVID-19 pandemic may have increased the risk and burnout in these professionals [28,29]. The COVID-19 pandemic has reduced human resources in ICUs; due to the daily overload felt in these units, nurses can feel physically and emotionally exhausted, discouraged, and, over time, lose motivation to work due to dissatisfaction and disillusionment [19,20,22–24,27,29,31,32]. Nurses experienced sudden and dramatic challenges on a personal level (e.g., moving away from family, changing houses) and in the workplace (e.g., increased workload, the threat of infection, traumatic episodes and feelings of frustration related to the death of patients). This work overload may have contributed to the increase in burnout among ICU nurses, depleting their emotional capacity to deal with the situations presented and the increase in daily demands [19,20,22–24,27,29,31,32].

It needs to be pointed out that organisational factors may have played a relevant role in the increase in burnout symptoms in ICU nurses. The already stressful working conditions and the escalation in work tasks and the change in the organisational environment may have directly affected the morale of the ICU nurses, combined with a feeling of loneliness and frustration, such as the lack of the social recognition of the work of these health professionals, generally resulting in challenging situations [23,28,31,32].

It is recommended that hospital administrations hire health professionals, mainly nurses, as a strategic and operational management strategy to reduce the risk of increased burnout during pandemic outbreaks [21,33]. Implementing evidence-based management strategies that demonstrate the effectiveness of break times, the number of working hours and the psychological and social support of these professionals is equally important. Health services, namely health managers and, especially in public systems, policymakers must progressively focus their attention on ICU nurses, offering them adequate support that the COVID-19 pandemic has aggravated. Thus, to minimise the increase in burnout during pandemic outbreaks, nurses' feelings of control over their schedules and tasks promotes their involvement in decision making, particularly in demanding situations such as those generated by pandemics.

This scoping review has limitations, like other studies, and the number of databases could be more significant. The languages considered may have contributed to the inaccessibility of essential studies in the study context. The fact that the COVID-19 pandemic timeline has been shortened, and that there are still studies to be carried out in the following context that are not yet available, may have made it difficult to understand and discuss the topic. To minimise this impact, the search strategy was specific but comprehensive, considering the primary databases in health sciences. However, this study may offer insights into future research on burnout in nurses working in the ICU, namely guidance for experimental studies with concrete measures to mitigate burnout. The intervention only focuses on the impact of the COVID-19 pandemic on ICU nurses, so it does not discuss other implications in other workplaces where nurses suffer the strains of the pandemic. Another limitation is that the scoping review only focused on nursing burnout. Hence, we could have missed some other issues that could have enhanced nurses' burnout, and other problems have not been discussed in this review.

## 5. Conclusions

The results of this scoping review point out that burnout among nurses working in the ICU is a reality today, which is why it is essential to implement strategies conducive to treating these repercussions. It will be equally important to guide strategies to prevent the appearance of this syndrome in these professionals, thus avoiding turnover in the ICU related to burnout, such as the implementation of programs in the respective services aimed at reducing stress, which may include the adoption of coping strategies, and it will be essential to involve managers in the implementation of leadership models that consider

professional recognition, personal appreciation and economical aspects. A contribution to these changes could be the elaboration of a list of potential factors contributing to the increase or decrease in the risk of burnout among ICU nurses. These will be good starting points for planning interventions to reduce the rise in burnout in ICU staff during pandemic outbreaks.

The originality of the present study can contribute to future investigations and changes in the working conditions of nurses in the ICU. The insights from this study will provide the necessary knowledge for developing interventions to address the increase in burnout in ICU nurses associated with pandemics.

**Author Contributions:** Conceptualization: A.L., M.T.M., C.F., M.S.F., M.F., J.T., M.S., V.P. and A.C. Validation A.L., M.T.M., C.F., M.S.F., M.F., J.T., M.S., V.P. and A.C. Writing—initial draft preparation: A.L., M.T.M., C.F., M.S.F., V.P. and A.C. Writing—review and editing A A.L., M.T.M., C.F., M.S.F., M.F., J.T., M.S., V.P. and A.C. All authors have read and agreed to the published version of the manuscript.

**Funding:** This research received no external funding.

**Institutional Review Board Statement:** Not applicable.

**Informed Consent Statement:** Not applicable.

**Data Availability Statement:** Not applicable.

**Acknowledgments:** The authors wish to thank Fernando Pessoa University.

**Conflicts of Interest:** The authors declare no conflict of interest.

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
