# Peer review of "The Burnout of Nurses in Intensive Care Units and the Impact of the SARS-CoV-2 Pandemic: A Scoping Review"

_nursrep, doi:10.3390/nursrep13010022_

Round 1

Reviewer 1 Report

General Comment

This is a relevant topic. It helps create a reflection of what could have caused burnout among intensive care nurses during the SARS-COV-2 pandemic. This is a relevant topic as it prepares the profession to understand its loopholes and areas for improvements.

Specific Comments

Abstract

At the end of the abstract please mention the limitations of the study. It is always professional to recognize your study limitations. No single study is limitations proof

Introduction

Well written.

You mention this in the section (According to Maslach and Leiter, burnout arises due to overwork, prolonged work 54 periods, low wages, interprofessional conflicts, work overload, lack of human and mate- 55 rial resources and even professional disappointment). Did the authors provide any statistics and figures on the contributions of these different factors to burnout? If yes, please add them here.

Materials and Methods

·         You have a heading for inclusion. You also must have a separate heading for exclusion or combine them into one heading ‘inclusion and exclusion criteria’

Results and discussion

·         Well written

·         Before the conclusion, add a limitation heading to acknowledge the study shortcomings

Well done

Author Response

Thank you very much for the comments and suggestions provided. We are sure that they have improved the quality and understanding of the work carried out by the reader.

  • Comment: At the end of the abstract please mention the limitations of the study. It is always professional to recognize your study limitations. No single study is limitations proof
  • Author response: 

    Given the reduced number of words to consider in the abstract, we can't add the limitations, but we add additional information in line 271 of the document

  • Comment: You mention this in the section (According to Maslach and Leiter, burnout arises due to overwork, prolonged work 54 periods, low wages, interprofessional conflicts, work overload, lack of human and mate- 55 rial resources and even professional disappointment). Did the authors provide any statistics and figures on the contributions of these different factors to burnout? If yes, please add them here.
  • Author response: As this is a scoping review, we do not consider it necessary to report these percentages, although they are identified when they exist in each article in table 2.
  • Comment: 

    You have a heading for inclusion. You also must have a separate heading for exclusion or combine them into one heading, ‘inclusion and exclusion criteria.’

  • Well written
  • Before the conclusion, add a limitation heading to acknowledge the study's shortcomingss
  • Arthor Response: Information has been added as requested

Reviewer 2 Report

Burnout arises due to overwork, prolonged work periods, low wages, work overload, interprofessional conflicts, lack of human and material resources and even professional disappointment. Then it makes sense to study the incidence of burnout syndrome in nurses who develop their practices in the ICU at the emergence of the SARS-CoV-2 pandemic. In fact, they are subject to physical and psychological overload, resulting in insomnia, physical fatigue, irritability, anxiety and it may culminate in pathologies of a depressive nature. Thus, the design of strategies aimed at nurses' health, which is subject to burnout during the pandemic by coronavirus was of interest.

The acronym ICU is presented in the abstract (background) without prior definition. To clarify the text to the reader it should be necessary to expose the meaning from the beginning. Some spell check is also required when the authors repeat sentences: "As such, all these factors induce high levels of stress, physical and psychological fatigue, As such, all these factors induce high levels of stress and physical and psychological fatigue and may result in exhaustion". Exactly the same when the authors write inadequately "Russia (n=1), =1)".

In the screening, 189 articles are excluded. Considering that the total number of articles identified is 229, the assessment of the reviewers should be made explicit. The exclusion criteria ought to be made clear. To recommend that hospital administrations hire nurses as a strategic and operational management strategy to reduce the risk of increased burnout during pandemic outbreaks is a truism that does not deserve prior research. Conclusions should go further, gathering and collecting the most important contributions of the articles analyzed.

Author Response

Thank you very much for the comments and suggestions provided. We are sure that they have improved the quality and understanding of the work carried out by the reader.

  • Comment: Burnout arises due to overwork, prolonged work periods, low wages, work overload, interprofessional conflicts, lack of human and material resources and even professional disappointment. Then it makes sense to study the incidence of burnout syndrome in nurses who develop their practices in the ICU at the emergence of the SARS-CoV-2 pandemic. In fact, they are subject to physical and psychological overload, resulting in insomnia, physical fatigue, irritability, anxiety and it may culminate in pathologies of a depressive nature. Thus, the design of strategies aimed at nurses' health, which is subject to burnout during the pandemic by coronavirus was of interest.
  • Author Response: Thanks for the feedback
  • Comment: The acronym ICU is presented in the abstract (background) without prior definition. To clarify the text to the reader it should be necessary to expose the meaning from the beginning. Some spell check is also required when the authors repeat sentences: "As such, all these factors induce high levels of stress, physical and psychological fatigue, As such, all these factors induce high levels of stress and physical and psychological fatigue and may result in exhaustion". Exactly the same when the authors write inadequately "Russia (n=1), =1)".
  • Author Response: Changes done as requested.
  • Comment: In the screening, 189 articles are excluded. Considering that the total number of articles identified is 229, the assessment of the reviewers should be made explicit. The exclusion criteria ought to be made clear. To recommend that hospital administrations hire nurses as a strategic and operational management strategy to reduce the risk of increased burnout during pandemic outbreaks is a truism that does not deserve prior research. Conclusions should go further, gathering and collecting the most important contributions of the articles analyzed
  • Author Response: Information added as requested